# Areas of Crime in Cities: Case Study of Lithuania

**Giedrė Beconytė [1],\*** , **Kostas Gružas [1]** and **Eduardas Spiriajevas [2]**

1    Institute of Geosciences, Vilnius University, M. K. Čiurlionio 21, 03218 Vilnius, Lithuania;
     kostas.gruzas@chgf.vu.lt
2    Centre for Social Geography and Regional Studies, Klaipėda University, S. Nėries g. 5,
     92227 Klaipėda, Lithuania; eduardas.spiriajevas@ku.lt
*    Correspondence: giedre.beconyte@gf.vu.lt

**Abstract:** In all countries, cities and their suburbs are the most densely populated areas. They are also the places visited by the largest number of tourists and one-day visitors, who inevitably run the risk of becoming victims of crime. It is, therefore, important, not only at national but also at the international level, to know the structure of urban crime and identify urban areas that differ in terms of their criminogenic situation. This requires a geographical approach and regionalisation based on the quantitative data that can offer it. This paper presents the results of a study using big data regarding violent crime, property crime and infringements against public order registered by the police in 2020 in the territories of three major Lithuanian cities and their suburbs (n = 149,239). Events in open spaces were separately addressed. A series of experiments were carried out using several spatial clustering methods. The automatic zoning procedure method that gave the best statistical results was then tested with different combinations of parameters. In each city, seven types of areas of urban crime were identified. Maps of crime areas (regions) were created for each city. The results of the regionalisation have been interpreted from a socio-geographical point of view and conform with previous sociological urban studies. Seven types of areas of crime have been identified, which are present in all the cities studied and, according to a preliminary assessment, roughly correspond to the socio-demographic and urban zones of each city. The maps of crime areas can be applied for crime prevention planning and communication, real estate valuation, strategic urban development planning and other purposes.

**Keywords:** crime; regionalisation; mapping; cities; urban crime; areas; regions

## 1. Introduction

Crime maps are compiled and used in order to understand the spatial patterns of the distribution of crime.

Crime maps are important tools for understanding the criminogenic situation in a given area and planning prevention and response measures. They are also used to compare the situations in different territories and monitor dynamics.

For convenience, we classify crime maps into several types:

(a)    *Detailed inventory maps* providing precise (address/point-level) information on events (Figure 1a). This is usually sensitive data, as known facts about some types of events may reduce the value of the property or, in sparsely populated areas, even be linked to individual persons, thus violating their right to privacy.

(b)    *Generalised inventory maps*, which provide data on the density and/or frequency of events at a spatial resolution that does not allow the identification of a precise location or person (Figure 1b). The aggregation can be performed in various ways, such as based on statistical grids, administrative units or clusters or by creating a statistical density surface (in this case, the data are further interpolated between points, thus creating certain distortions).

(c)   *Analytical maps*, where derived characteristics such as density, hotspots, etc., are presented. Analytical maps help to identify patterns and trends (Figure 1c). In such maps, crime data are often linked to indicators that may influence the distribution of crime.

(d)   *Regionalisation maps*, which divide the territory into a relatively small number of contiguous areas that differ in terms of certain characteristics, i.e., so-called regions or areas of crime (Figure 1d).

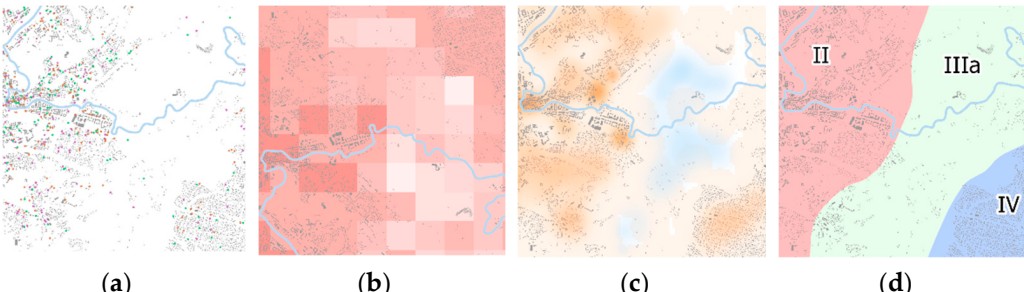

**Figure 1.** Fragments of crime maps: detailed inventory (**a**); 1 km × 1 km gridded inventory (**b**); analytical (hotspot) (**c**); regionalisation, Roman numerals refer to different regions (**d**).

Spatial zoning or regionalisation maps have traditionally been used for decision-making in a wide range of fields [1–4]. On one hand, the thematic information that they contain is highly generalised, which makes them easy to understand for national and local politicians, development strategists and urban planners and managers, who are responsible for making decisions that are important for the entire population of the country or in particular geographical regions [5–8]. On the other hand, a region is much more than a simple aggregation of spatial data. Regions are distinguished on the basis of a number of interrelated criteria and the contiguity of spatial units that are similar in terms of these criteria [9,10]. The characteristics of the regions must be described and interpreted in a meaningful way within a social–geographical context. Only then will a regionalisation be useful for decision-makers [11–13].

The term "region" in this article is distinct from an administrative, ethnic or other formally recognised region. As the term "region" is more commonly used to describe large complexes of natural, political or ethnic features, where appropriate, we also use the more general term meaning essentially the same thing: "area". We define *area of crime* as a geographically coherent region within a city characterised by a particular crime pattern.

The aim of this study is to select appropriate methods and parameters of regionalisation to distinguish the areas of crime in Lithuanian cities according to the patterns of crime contained in the large volume data in the police's register of incidents. Once the logical and explainable regions have been identified, a map can be created and used for further insights and analysis.

The following research questions we considered:

- Which of the known spatial clustering methods achieves the best validity for given country-specific crime data?;
- Is it possible to identify districts that are consistent with a common understanding of the characteristics of urban areas?

The specific aim of this study was to delineate the areas of the three largest Lithuanian cities according to the structure of crime, especially in less densely populated urban areas, where alternative crime-prevention measures may be beneficial.

Previous studies have found clear differences in the spatial distribution of crime between densely populated areas (cities) and rural areas. Cities have higher overall risk of crime. This is in the first instance due to the higher population density and increased opportunities for criminal activities. However, there are also differences in the relative

number of incidents and the structure of incidents—for example, cities in Lithuania have relatively fewer violent crimes and more thefts.

Vilnius, Kaunas and Klaipėda are the three largest cities in Lithuania and serve as important economic, political and cultural centres in the country. As Lithuania has transitioned from a post-Soviet state to a modern European country, all three cities have witnessed significant economic growth and diversification, the development of infrastructure and tourism, the construction of new residential areas and the development of commercial zones and green spaces. They are now attractive, dynamic cities with a consistently improving quality of life. These are also the cities with the most complex crime patterns, which, if understood, would offer prospects for more effective prevention strategies.

While the three cities share many common characteristics, they also exhibit distinct socio-demographic, economic and political differences. Vilnius is about twice the size of Kaunas and Kaunas is twice as large as Klaipėda. These cities belong to differing ethnographic regions and have undergone different historical developments.

Vilnius stands out as the capital and the largest city in Lithuania, with a population of almost 0.6 million. It is also the most ethnically diverse city in Lithuania. It has a higher proportion of foreign residents and expatriates due to its role as the capital and cultural hub. Vilnius is the economic heart of Lithuania, being home to a wide range of businesses, financial institutions and government offices. The city has a highly developed service sector, including IT, finance and tourism.

Kaunas is the second-largest city in Lithuania, with a population of over 300,000 people. It has a more homogeneous ethnic Lithuanian population compared to Vilnius. Kaunas has a strong historical and cultural identity, with a focus on preserving Lithuanian traditions and heritage. The city's lower cost of living and well-developed infrastructure make it attractive for businesses, both local and international.

Klaipėda, as a coastal city, plays a crucial role in maritime and trade-related activities, contributing to its unique economic and demographic profile. It is the country's third-largest city, with a population of around 150,000 people. It has a diverse population due to its port and international connections, as well as historical factors.

In the text that follows, "Vilnius", "Kaunas" and "Klaipėda" refer to the combined territory of the city itself and its suburbs, while "Vilnius city", "Kaunas city" and "Klaipėda city" only refer to the city's municipal territory.

In all cities, the distribution of police-recorded incidents is clearly heterogeneous. However, the differences, especially in the non-central parts of the cities, are quite subtle and difficult to identify. This is where regionalisation can help.

The present study is original in several respects:

(a) The regionalisation is carried out in cities rather than in large areas. It is a pilot study with the aim of contributing to better crime management in densely populated areas.
(b) The variables of the regionalisation are primarily related to the structure of crime, distinguishing between open spaces and private/semi-private spaces.
(c) Aggregated data divided into five classes are used for regionalisation. This reduces distortions due to very small or occasionally very large values.

## 2. Previous Research

The history of the cartography of crime, spanning almost two centuries, has taken us from isolated mapping techniques to sophisticated analyses and design of instruments for reducing the risk of crime. For a more detailed overview of previous crime mapping research and for the arguments underlying the importance of national data-driven studies, see our previous publication [14].

In Lithuania, systematic research into spatial criminology began in the second decade of the 21st century via the study of small territories such as parts of towns [15,16]. As the availability of spatial data improved, data from the Lithuanian Police Register of Recorded Events (hereinafter RERP) began to be used, and inventory maps were developed. From 2015 to 2019, datasets from the RERP, covering the whole country, were collected

in an identical structure. Large volumes of the RERP data containing millions of records have been made available for research, where they have been aggregated using different methods. Analytical maps have been produced. Since 2020, the structure of the RERP data has changed. In this year, the COVID-19 pandemic started, leading to a significant change in the spatial distribution of events. Currently, RERP data from 2020 onwards are used for the studies.

In our previous papers [14,17], we provided a more detailed summary of geographical crime research in Lithuania. These studies also include the inventory and analytical maps of the crime situation. These maps were used in the research presented in this paper—both for initial insights and making assumptions about the characteristics of crime, which actually shape the regions.

The experiments, conducted by one of the authors between 2021 and 2023 [18], included a detailed analysis of the crime data for Vilnius city, the identification and description of the hotspots and the socio-demographics of their inhabitants, as well as the identification of five intervals of the total number of events. In this paper, we present the results of the first attempt at the coherent criminological regionalisation of Lithuanian cities.

Statistical interpretation of the regionalisation results alone is not sufficient. Simply reducing the overall error may result in a large number of small or highly fragmented districts, which do not add value compared to conventional statistical grids or density maps.

Spatial studies in cooperation with criminologists have provided a good understanding of the overall dynamics of crime in Lithuania, as well as a rough estimate of the peculiarities and differences between urban and rural areas and between areas of urban crime. Thanks to the knowledge gained, we can critically assess and interpret the results of complex quantitative studies. This makes it possible to carry out regionalisation on the basis of RERP data and assess its suitability not only statistically but also in the context of criminological research.

In the scientific literature, it is possible to find various studies of the application of methods of regionalisation, with the common goal of describing a complex spatial phenomenon in a small number of spatially contiguous regions (preferably several), in such a way that the characteristics of the phenomenon are as similar as possible within a region and as different as possible between regions. Regionalisation methods can be sophisticated, involve the user to varying degrees (e.g., requiring the input of the number of regions or of pre-defined similarity characteristics) or identify regions from data alone [19]. A comparison of methods of agglomerative clustering, namely *SKATER*, *REDCAP*, *AZP* and *Max-P-Regions*, is provided by Lattimer and Lattimer [20].

For the present study, we relied on a meta-analysis that has identified the essential features of all regionalisation methods [21]. This study also provides an overview and comparison of eight groups of supervised regionalisation methods. We also analysed studies that used spatial clustering methods for the analysis of crime and similar phenomena: constructing regions for homicide research using the REDCAP method and the model-based clustering of expectation–maximisation and K-means algorithms in crime hotspot analysis [22].

Based on the review of the literature, we can state that the most frequently cited methods that are suitable for the crime regionalisation task are agglomerative clustering, *SKATER* (spatial cluster analysis by tree edge removal) [23,24], *SKATER-CON* [25], *REDCAP* (regionalisation with dynamically constrained agglomerative clustering and partitioning) [26,27], *AZP* (automatic zoning procedure) [21,28,29], *Max-P-Regions* [30], *GeoK-Means* [31], *DBSCAN* (density-based spatial clustering of applications with noise) [32] and *SCHC* (spatially constrained hierarchical clustering) [33,34].

Regions can further be used for better understanding the phenomenon, making spatial and temporal predictions and planning with an eye to better problem management [35–38]. While generally acknowledging the applied value of regional mapping results, most authors emphasise that there is no one-size-fits-all approach and results may vary depending on the method used or its parameters [38,39]. Therefore, as described below, we carried out

experiments (It is important not to confuse the term "experiment" as used in this article with the experimental methods in criminology that use randomised controlled trial. In our study, experiments were carried out with the aim of selecting appropriate regionalisation methods and their parameters) using several methods of spatial clustering.

### 3. Materials and Methods

*3.1. Experiment Design*

This study consisted of the following steps:

1. Systematising insights from the literature and previous regionalisation experiments.
2. Preparing data for analysis:
   a. Geocoding and cleaning of RERP crime data;
   b. The aggregation of point-event data in a grid of 500 m × 500 m cells that were further used as inputs for spatial clustering;
   c. Adding summarised population data;
   d. The reclassification of event count values;
   e. The identification of urban and suburban areas to be investigated.
3. Experimental selection of a regionalisation method:
   a. The selection of the initial number of clusters;
   b. The application and evaluation of different spatial clustering methods for each city;
   c. Experiments changing the number of clusters via the selected methods;
   d. The selection of the most suitable clustering method and the number of clusters.
4. Regionalisation and mapping.
5. Analysis and interpretation of the outcomes.

*3.2. Data*

The data used for the study comprised the 2020 data from the Lithuanian Police Register of Recordable Events (RERP) (Table 1). The 2020 data were chosen because they were the most recent available data at the time of this study. In addition, the data reflect the specific situation of the COVID-19 pandemic and the quarantine period and cover the whole year. Three event types were chosen as regionalisation variables:

- Violent crime (VIO), such as homicide, murder, assault, manslaughter (bodily injury), sexual assault, rape, robbery, abduction and harassment;
- Property offences (PRO), such as theft, destruction or damage to property;
- The infringement of public policy (IPP), such as breaches of the peace, illicit consumption of alcohol, noise, littering, etc.

**Table 1.** Number of RERP events investigated in cities by event type.

| City (with Suburban Territories) | Area, km$^2$ | Population, Thousand | Types of Events | | | |
|---|---|---|---|---|---|---|
| | | | *VIO* | *PRO* | *IPP* | *Total* |
| *Vilnius* | 737 | 624 | 19,488 | 26,036 | 35,753 | *81,277* |
| *Kaunas* | 478 | 372 | 10,770 | 15,382 | 16,728 | *42,880* |
| *Klaipėda* | 289 | 170 | 6268 | 6931 | 11,883 | *25,082* |
| *Total* | *1504* | *1166* | *36,526* | *48,349* | *64,364* | **149,239** |

The choice of the regionalisation variables was made on the basis of previous studies, which show that there are considerable spatial differences in their dispersion in Lithuanian cities, and the ratio of VIOs to PROs markedly differs in sparsely and densely populated areas [14,40]. In addition to these three types, the RERP database includes traffic accidents, drug possession and distribution accidents and unclassified ("other") accidents. Traffic accidents can only be indirectly linked to criminal behaviour, and they depend more on

the characteristics of the road and street network than geographical location. Drug-related events are relatively few in number (5–10 times fewer than the events used in the analysis) and clearly concentrated, and they would, therefore, have a predictable influence on the identification of regions. The variety of 'other' events is very large, and most of them do not have a clear link with criminal behaviour. For these reasons, these types of events have not been used for regionalisation.

Our previous research shows that the share of open space events is a potential region-shaping factor in Lithuania. For example, the gradual decrease in the number of events in open spaces from 2015 to 2019 (45%) was relatively larger than of those in premises (14%); the growth of open-space crime (that is opposite to overall trend) is solely concentrated in Kaunas [14]. In Vilnius city, the ratio of the number of violent incidents in open spaces varied between different areas during the COVID-19 pandemic [40]. Therefore, in this study, we also grouped all three types of events into the following groups: events in open spaces (open yards, streets, squares, parks, etc.) and events in private/semi private spaces (residences or other premises, enclosed yard, etc.).

In the regionalisation experiments, point-event data summarised in rectangular grids were used. It is obvious that the smaller the grid, the more accurate the results of the analysis, but in the case of events whose locations were not precise, the excess precision would not add value [41–44] and only complicate the regionalisation. Previous studies have shown that 500 m × 500 m grids are sufficient to analyse the population density of Lithuania and draw reasonable conclusions [45]. The dispersion of the above types of crime events is closely linked to population density. Therefore, it would not make sense to investigate crime with a higher spatial precision. Thus, 500 m × 500 m cell grids were also used in this regionalisation study.

The RERP event data were geocoded, the outliers were removed, and counts (absolute values) for all three event types were calculated for each grid cell. It is commonly believed that crime is closely linked to the size of the population in a given area [46,47]. Based on this assumption, we included population data as one of the factors for regionalisation.

Crime surveys are complicated by the fact that the data are not normally distributed, their spatial structure is complex and the variation in values is high. Applying spatial clustering methods described in the literature on a grid with absolute numbers yielded very highly fragmented regions. Therefore, we decided to categorise the number of events and population in the grid cells prior to regionalisation. For ease of interpretation, 6 categories have been established, corresponding to the conventional scores of "none" (assigned a value of 0), "a low number" (1st to the 20th percentile of crime events for each individual city, assigned a value of 1), "less than average" (21st to 40th percentile, assigned a value of 2), "average" (41st to 60th percentile, assigned a value of 3), "more than average" (61st to 80th percentile, assigned a value of 4) and "a high number" (above the 80th percentile, assigned a value of 5). Seven attributes were categorised for the three cities, i.e., 21 sets of categories were obtained. In six cases, the values of the 20th and 40th percentiles coincided and were equal to 1. In these cases, all cells with a single event were assigned to the first category, while the second category was not assigned any cell. The percentile values for all attributes are presented in Table 2. The set of parameters used for regionalisation is shown in Table 3.

The municipal boundary is not an objective criterion for distinguishing between homogeneous areas of crime. On the contrary, previous studies have shown that areas close to the city boundary can have a very similar pattern of crime [48,49]. Therefore, we decided to include the suburban territories in the study. In Lithuania, the strict criteria and methodology for identifying urban suburbs have not been defined, so, for this study, we developed a methodology for their identification based on the assessment of similarity to the adjacent urban area. Selected non-urban elderships (smallest administrative units) adjacent to cities were covered with a grid of 100 m × 100 m cells. To assess similarity, the criteria of population density, density of the road and street network, percentage of the built-up area and spatial connectivity were considered. The values found for these criteria

were compared to the equivalent statistics for the areas of the specific city whose suburbs were identified. The methodology for distinguishing suburbs is rather complex, and the result—the boundaries of a spatially coherent area adjacent to a city—is not central to the regionalisation exercise. We, therefore, decided not to go into further detail about this issue in this paper. It is sufficient to say that the areas of Vilnius, Kaunas and Klaipėda with their suburban territories were defined. These areas were further regionalized (Figure 2).

**Table 2.** Attribute percentile values for all cities.

| City | Attribute | Percentile Values | | | | Max Value |
|------|-----------|------|------|------|------|-----------|
| | | **20th** | **40th** | **60th** | **80th** | |
| *Vilnius* | *VIO_NOT_OPEN* | 1 | 2 | 5 | 14 | 235 |
| | *PRO_NOT_OPEN* | 1 | 2 | 4 | 16 | 440 |
| | *IPP_NOT_OPEN* | 1 | 2 | 5 | 17.5 | 528 |
| | *VIO_OPEN* | 1 | 1 | 2 | 5 | 84 |
| | *PRO_OPEN* | 1 | 1 | 3 | 8 | 88 |
| | *IPP_OPEN* | 1 | 1 | 3 | 10 | 123 |
| | *POP* | 10 | 29 | 79 | 227 | 5531 |
| *Kaunas* | *VIO_NOT_OPEN* | 1 | 2 | 5 | 14 | 183 |
| | *PRO_NOT_OPEN* | 1 | 2 | 4 | 14 | 259 |
| | *IPP_NOT_OPEN* | 1 | 2 | 5 | 15 | 230 |
| | *VIO_OPEN* | 1 | 1 | 3 | 7 | 45 |
| | *PRO_OPEN* | 1 | 2 | 3 | 7.5 | 60 |
| | *IPP_OPEN* | 1 | 2 | 3 | 10 | 155 |
| | *POP* | 9 | 32 | 94 | 295 | 5010 |
| *Klaipėda* | *VIO_NOT_OPEN* | 1 | 2 | 4 | 22.5 | 246 |
| | *PRO_NOT_OPEN* | 1 | 1.5 | 3 | 18.5 | 330 |
| | *IPP_NOT_OPEN* | 1 | 2 | 4 | 34 | 235 |
| | *VIO_OPEN* | 1 | 2 | 4 | 12 | 111 |
| | *PRO_OPEN* | 1 | 1 | 3 | 11 | 84 |
| | *IPP_OPEN* | 1 | 2 | 3 | 19.5 | 201 |
| | *POP* | 8 | 21 | 56.5 | 193.5 | 5494 |

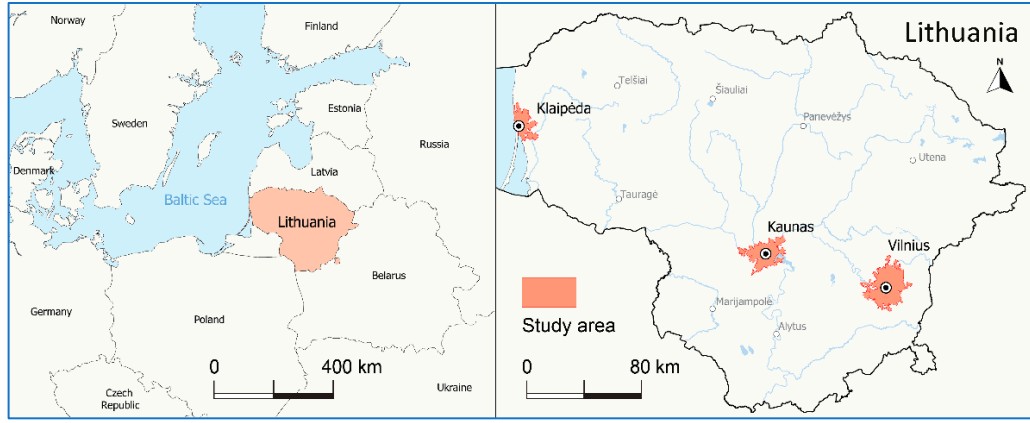

**Figure 2.** Locator map and areas under investigation: Vilnius, Kaunas and Klaipėda, including their suburbs.

**Table 3.** Attributes of the dataset used for spatial clustering.

| Attribute | Values | Description |
|---|---|---|
| GRID_ID | Text | Unique identifier of the cell. |
| VIO_NOT_OPEN_CAT | Integer, [0..6] | Category of violent crime in non-open spaces for the cell. |
| PRO_NOT_OPEN_CAT | Integer, [0..6] | Category of property crime in non-open spaces for the cell. |
| IPP_NOT_OPEN_CAT | Integer, [0..6] | Category of infringement of public policy in non-open spaces for the cell. |
| VIO_OPEN_CAT | Integer, [0..6] | Category of violent crime in open spaces. |
| PRO_OPEN_CAT | Integer, [0..6] | Category of property crime in open spaces. |
| IPP_OPEN_CAT | Integer, [0..6] | Category of infringement of public policy in open spaces. |
| POP_CAT | Integer, [0..6] | Category of population size for the cell. |

*3.3. Selection of Spatial Clustering Method*

As mentioned in Section 1, different regionalisation methods and variations in their parameters may produce significantly different results. Therefore, experiments were carried out using the four regionalisation methods that have yielded the best statistical estimates in a series of experiments on crime data for the entire territory of Lithuania (Gružas, 2023): AZP, REDCAP, SCHC and SKATER. All four methods are described in detail in the *GeoDA* software documentation [50].

- AZP: the automatic zoning procedure (AZP) uses heuristics to find the best set of combinations of contiguous spatial units into p regions, minimizing the within sum of squares as a criterion of homogeneity.
- SCHC: Spatially constrained hierarchical clustering algorithm. This method builds up the clusters using the following agglomerative hierarchical clustering methods: single linkage, complete linkage, average linkage and Ward's linkage (a special form of centroid linkage).
- SKATER: the divisive hierarchical clustering algorithm based on the optimal pruning of a minimum spanning tree into several clusters, seeking the maximal similarity of the values of the selected variables while retaining spatial contiguity.
- REDCAP: Regionalisation with dynamically constrained agglomerative clustering and partitioning. It builds a spanning tree in 4 different ways (single linkage, average linkage, Ward's linkage and complete linkage). It also provides 2 different ways (first-order and full-order constraining) of pruning the tree to find clusters.

None of these methods require predefined characteristics of similarity.

The initial number of clusters was based on a previous study by one of the authors [16], which identified six territorial clusters of crime in Klaipėda city. Starting with six clusters was also a logical choice for Vilnius: previous studies [18] have observed at least six variations in terms of the absolute and relative number of incidents registered by police, as well as of the ratio of theft to violent crime.

The number of potential crime clusters depends on the socio-demographic and urban territorial structures of the analysed cities. In the major cities of Lithuania, we can differentiate between the districts of the historical part of the city (I); the residential districts formed during the Soviet era (II); the industrial or formerly industrial areas (III); the private residential areas within the city boundaries (IV); the inhabited suburban areas that were previously rural (V); the former adjacent small towns that are experiencing urban regeneration processes with developing road networks and new business entities, which are linked to the city by the daily commuting of the inhabitants to the central areas (VI); and green urban areas, such as parks and water resources (banks of rivers, lakes, ponds, rivers and

rivulets) designated for recreational use by local residents (VII) [16]. Considering the size of each of the cities, as well as their spatial diversity and the fact that some structures might have been overlooked in the visual assessment, we also carried out experiments with seven and eight clusters.

The results of clustering were evaluated based on the of internal cohesion and external separation of the regions. The between-sum-of-squares (BSS)/total-sum-of-squares (TSS) ratio was used as a criterion to assess the results, as it quantifies the proportion of the total variability in the data that is explained based on the spatial clustering and provides insight into how well the regionalisation method captures the underlying spatial structure in the data:

$$BSS/TSS = (BSS)/(TSS) \tag{1}$$

where

$$BSS = \Sigma n_i \times (M_i - M)^2 \tag{2}$$

$$TSS = \Sigma_i \, \Sigma_j \, (X_{ij} - M)^2 \tag{3}$$

$n_i$ is the number of observations in group I, $M_i$ is the mean of group I, M is the overall mean of the entire dataset and $X_{ij}$ is the individual data point.

BSS represents the variance between the clusters. TSS represents the total variance in the dataset. A higher BSS/TSS ratio indicates that a larger proportion of the total variation is explained by the spatial clusters, suggesting that the regions are significantly different from each other. A higher BSS/TSS ratio indicates that the clustering method has successfully grouped similar values together, leading to more distinct spatial clusters. Ideally, the BSS/TSS ratio should approach 1. In practice, very high values cannot be achieved because of the spatial contiguity requirement and the many possible combinations of the seven variables. In our experiments, we assessed the characteristics of the identified cluster centres, the compactness of the regions and the relevance of the regions to the main urban areas. In most cases, BSS/TSS values greater than 0.6 represented sets of regions that could already be more or less consistently interpreted.

A series of experiments with different parameters and specified-at-the-outset number of clusters (6, 7 and 8) were performed for each city. The experiments used the Euclidean distance function and the queen contiguity. No transformation was used because the data were already aggregated and reclassified. Other parameters specific to individual methods (Table 4) were tested using *GeoDA 1.22* open-source software for exploring spatial patterns (https://geodacenter.github.io/, developed by Luc Anselin, accessed on 14 December 2023). The best BSS/TSS ratios are presented in Table 5. In both tables, the colours on the maps have no special meaning. The different colours merely indicate the different spatial clusters.

**Table 4.** Tested methods and their parameters.

| Method | Parameter | Values Tested | Best Result | Example of Regionalisation (Kaunas, 7 Clusters, Best Result) |
|---|---|---|---|---|
| *SCHC* | *Method* | Ward's Linkage Single Linkage Complete Linkage Average Linkage | Ward's Linkage |  |

**Table 4.** *Cont.*

| Method | Parameter | Values Tested | Best Result | Example of Regionalisation (Kaunas, 7 Clusters, Best Result) |
|---|---|---|---|---|
| *SKATER* | *Minimum Bound* | None Population: 100 to 1500 at intervals of 100 | None |  |
| | *Min region size* | None Grids: 10 to 60 at intervals of 5 | None | |
| *REDCAP* | *Method* | Full-Order Ward's Linkage First-Order Single Linkage Full-Order Complete Linkage Full-Order Average Linkage Full-Order Single Linkage | Full Order Ward's Linkage |  |
| | *Minimum Bound* | None Population: 100 to 1500 at intervals of 100 | No impact | |
| | *Min region size* | None Grids: 10 to 60 at intervals of 5 | None | |
| *AZP* | *Minimum Bound* | None Population: 100 to 1500 at intervals of 100 | 1000 |  |
| | *Min region size* | None Grids: 10 to 60 at intervals of 5 | No impact | |
| | *Arisel* | None, 10 to 100 at intervals of 10 | No impact | |
| *AZP-Tabu Search* | *Tabu length* | 10, 15, 25, 50, 100, 300, 600, 1200 | 1200 |  |
| *AZP-Simulated Annealing* | *Cooling Rate* | 0.70, 0.80, 0.85 | 0.70 |  |
| | *MaxIt* | 1, 3, 7 | 7 | |

**Table 5.** The best between-sum-of-squares/total-sum-of-squares ratios (BSS/TSS).

| Territory | Number of Clusters | Method | | | |
|---|---|---|---|---|---|
| | | *SCHC* | *SKATER* | *REDCAP* | *AZP* |
| *Vilnius* | *6* | 0.549 | 0.455 | 0.556 | 0.572 |
| | *7* | 0.560 | 0.467 | 0.564 | 0.603 |
| | *8* | 0.570 | 0.479 | 0.572 | 0.621 |
| *Kaunas* | *6* | 0.586 | 0.531 | 0.579 | 0.627 |
| | *7* | 0.597 | 0.543 | 0.595 | 0.640 |
| | *8* | 0.607 | 0.553 | 0.607 | 0.645 |
| *Klaipėda* | *6* | 0.679 | 0.620 | 0.679 | 0.700 |
| | *7* | 0.692 | 0.632 | 0.693 | 0.695 |
| | *8* | 0.702 | 0.642 | 0.703 | 0.699 |

In all cases, the best statistical results were obtained using AZP with Ward's linkage and, for Vilnius and Kaunas, the *Minimum Bound* of 1000 inhabitants. Increasing the *Minimum Bound* further, pairs of clusters, especially those with lower values in the population categories, became intermingled with each other, and no additional value was created for the interpretation. Increasing the number of clusters to 8 also led to an undesirable intermingling of clusters. This shows that a slightly better statistical estimate does not necessarily lead to better regionalisation (Table 6)

**Table 6.** The example of Vilnius. Less satisfactory results despite slightly better BSS/TSS estimates: while the overall picture was very similar, the configuration of regions (B) and (C) as much more complex and interspersed.

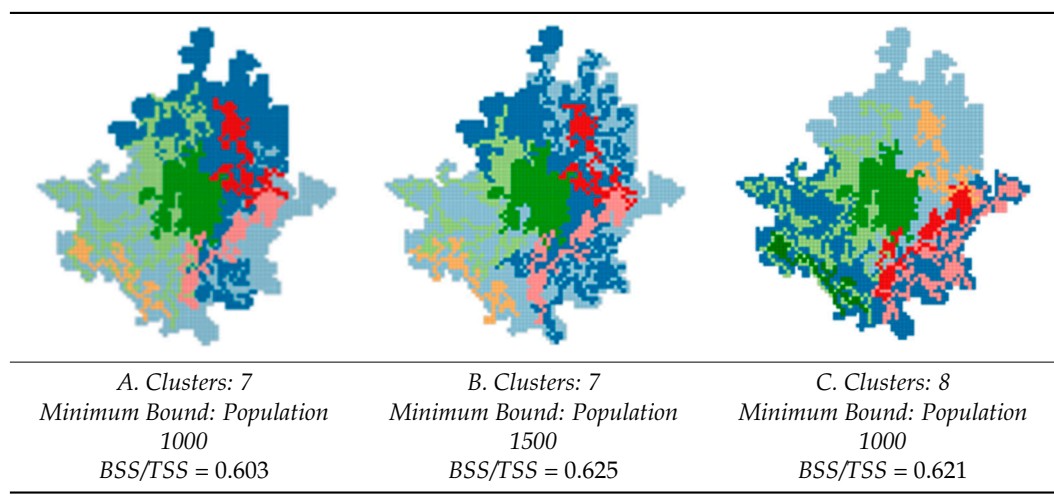

| *A. Clusters: 7* | *B. Clusters: 7* | *C. Clusters: 8* |
|---|---|---|
| *Minimum Bound: Population 1000* | *Minimum Bound: Population 1500* | *Minimum Bound: Population 1000* |
| *BSS/TSS = 0.603* | *BSS/TSS = 0.625* | *BSS/TSS = 0.621* |

We also performed a series of experiments via the AZP *Tabu Search* and *Simulated Annealing* methods with just 7 clusters, varying their parameters in the manner recommended in the literature. With some combinations of parameters, the BSS/TSS estimate increased (not substantially) compared to the original AZP (Table 7), but the proportions and configuration of regions became less acceptable, as can be seen in the Kaunas map examples in Table 4.

**Table 7.** AZP for 7 clusters: the best between-sum-of-squares/total-sum-of-squares ratios (BSS/TSS).

| Territory | AZP Heuristic | | |
| --- | --- | --- | --- |
| | *Original* | *Tabu Search* | *Simulated Annealing* |
| *Vilnius* | 0.572 | 0.629 | 0.677 |
| *Kaunas* | 0.627 | 0.584 | 0.693 |
| *Klaipėda* | 0.700 | 0.684 | 0.719 |

## 4. Results

The AZP method was applied to the data of the cities of Vilnius, Kaunas and Klaipėda and resulted in the emergence of seven spatial areas of urban crime in each city. Most regions are comparable between the cities and their known socio-demographic districts. However, cities have their own specificities, which is why we considered it appropriate to identify a total of eight areas of crime based on the identified cluster centres.

Profiles of the crime areas and approximate correspondence with the general characteristics of urban areas known from previous studies [16] are presented in the table below (Table 8). As the geodemographic classification has only been carried out for Klaipėda, the correspondence is only indicative. If geodemographic clusters were to be identified in the future in each city using the same method, the spatial coverage of the clusters could be quantitatively compared with the areas of crime.

**Table 8.** Tentative correspondence between the crime areas and urban areas of Vilnius, Kaunas and Klaipėda.

| Area of Crime | Characteristics of the Crime Area *Specificity of Individual Cities* | Corresponding Urban Socio-Demographic Type |
| --- | --- | --- |
| *I* | Average crime rate, with all types of crime being proportional to population. Highest number of VIO. *In Vilnius, a lower number of VIO events occur in open spaces than in the other two cities.* | Central (historical) parts: medium population density, densely built-up with some derelict buildings and prevalence of SMEs' commercial services. In Vilnius, this cluster also includes densely populated residential areas around the city centre. |
| *II* | Relatively high levels of crime, more in premises, larger proportion of IPP and PRO. *In Vilnius, more VIO events, both in open spaces and in premises. In Klaipėda, generally lower crime rate.* | Residential blocks of flats, densely built-up, high population density, commercial services of SMEs and large retail chains. Developed social infrastructure, such as schools, kindergartens, healthcare centres and interspersed green spaces. Intensive mobility of locals for daily needs around shopping centres. |
| *III* | Lower level of crime and predominance of crime on premises. *In Vilnius, lower rate of IPP in open spaces.* | Residential areas with detached houses within the city, medium population density, low level of commercialisation of the area and social control technologies (video surveillance, fences, gates used in the delineation of private property |
| *IV* | Large proportion of crimes in open spaces, especially VIO and IPP. *In Vilnius, relatively more VIO events, both in open spaces and on premises.* | Peripheral urban areas of mixed development (residential, medium and small business and manufacturing enterprises), lower than average population density and operational commercial structures. |

**Table 8.** *Cont.*

| Area of Crime | Characteristics of the Crime Area<br>*Specificity of Individual Cities* | Corresponding Urban Socio-Demographic Type |
|---|---|---|
| *V* | Low-to-average levels of recorded crime, of which crime on premises predominate.<br>*In Vilnius, higher rates of PRO and IPP.* | Single-family detached house areas in the suburbs and gated communities, with average and lower than average population densities. |
| *VI* | Relatively high number of VIO events, specifically in open spaces.<br>*In Kaunas, higher population density and high rates of PRO and IPP.* | Sparsely populated areas with derelict buildings. Areas are being regenerated, and businesses are operating. These areas are designated for economic activities rather than for permanent residence. |
| *VII* | Below-average levels of recorded crime. Relatively higher number of IPP events in open spaces.<br>*In Klaipėda, virtually uninhabited areas; very high relative crime levels, especially of IPP in open spaces, are due to extremely sparse populations.* | Settlements remote from the core urban area, which are within the economic pull of the metropolitan area due to the daily commuting of inhabitants. Average and below-average population density. |

The digital versions of the maps of urban crime areas are available online at https://vu-lt.maps.arcgis.com/apps/dashboards/ad061d31a59b4232861734f0628069c2. The printable versions are shown in the Figures 3–5 and are available for download at http://kc.gf.vu.lt/?page_id=2178#AZP_3cities (both links accessed on 14 December 2023).

Vilnius is characterised by a large and heterogeneous central area and a relatively high concentration of events in peripheral or suburban areas. The regions are of proportional size. Although the central area (I) could be further subdivided, the distinction between Type II, III and IV suburban areas is particularly valuable.

Kaunas has the highest density of incidents in the central part of the territory, but there are also areas of higher crime further away from the centre, especially in the southern and eastern residential areas, which are characterised by relatively higher levels of violent crime. In this city, the difference between the residential areas corresponding to areas of crime II and III is quite minor.

In discussions with sociologists, it was noted that Kaunas city's structures are very complex and particularly difficult to analyse. It is, therefore, interesting that the clusters of crime in this city are rather compact and relatively well separated.

In Klaipėda, crime is heavily concentrated in the central part of the studied area, with no pronounced hotspots located on the periphery or in the suburban areas, which means that sparsely populated districts of the city are generally very safe.

The example of Klaipėda in this study allows us to define one more possible area of crime (VIII): areas that are very sparsely populated and dominated by forested areas, Curonian lagoon coastal settlements and collective garden communities.

This is because Klaipėda's green spaces are not integrated into the city's urban areas but instead occupy large contiguous areas on the periphery. The integration of suburbs and green spaces into the urban fabric differs from that of other major cities, particularly due to the lower concentration of population and economic activities. Crime there takes place in public spaces, nearly always being committed by non-local people.

In this study, Vilnius and Kaunas do not have such a cluster, so the 'green' cluster of Klaipėda has been identified as area of crime VII, which is the most similar in terms of its characteristics. Considering a better solution for the regionalisation of Klaipėda, the extent of the area of crime VII could be smaller, especially in the southern part of the city, and next to it could be area of crime VIII, solely reflecting the structure of crime in the green spaces.

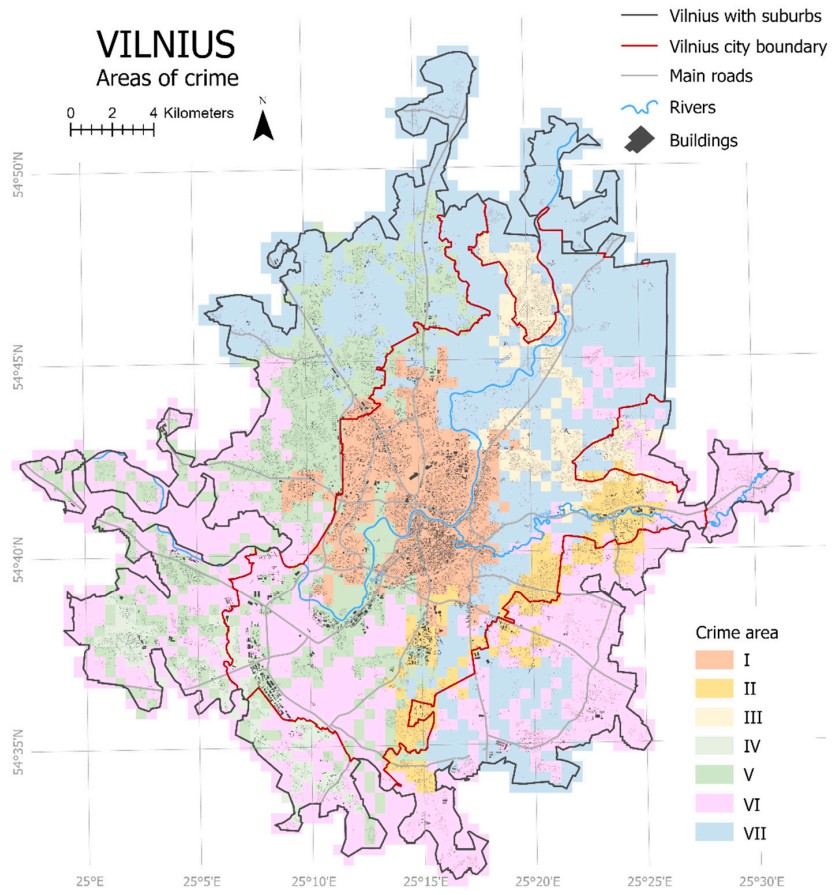

**Figure 3.** Areas of urban crime in Vilnius.

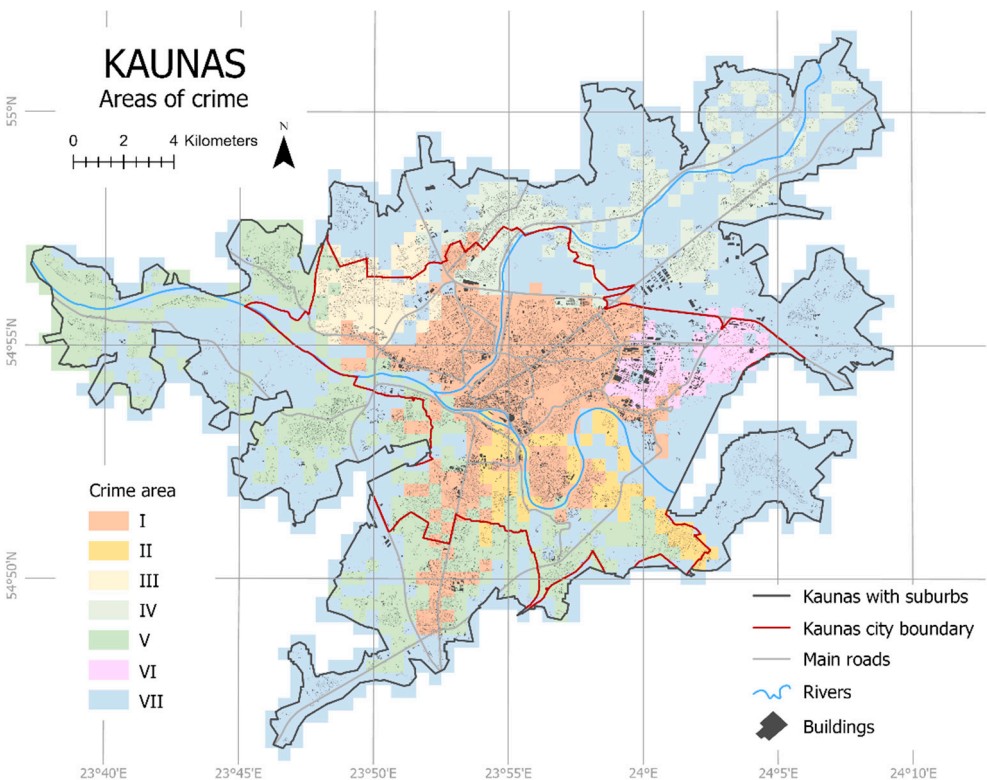

**Figure 4.** Areas of urban crime in Kaunas.

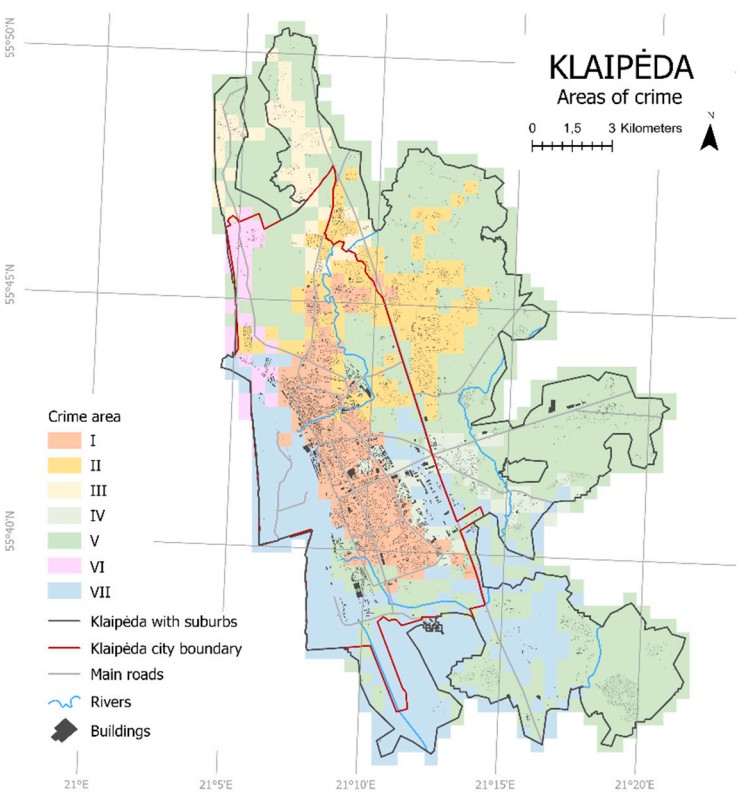

**Figure 5.** Areas of urban crime in Klaipėda.

## 5. Discussion and Conclusions

This study contributes to the field of regionalisation and urban crime research. The results of this study are seven types of areas of crime that reveal a specific crime pattern with a clearly defined central zone and a variety of different types of territory enveloping it. It has been found that, as the literature suggests, the number of incidents depends on the size of the population. However, similarly dense urban areas can be differentiated according to the structure of crime. The largest differences observed are related to the proportion of violent crime and the proportion of events in open spaces. Overall, the results are in line with crime concentration theories [51] and do not contradict the results of previous studies of crime in Lithuanian cities [14–16,52]. From a methodological point of view, it confirmed the suitability of the AZP method for the regionalisation of crime event data.

The spatial clustering methods tested allowed us to identify areas (regions) of crime with common features in the three largest Lithuanian cities, which generally correspond to the urban and socio-demographic structure of the cities and reflect the peculiarities of urban crime identified in previous studies. A series of more than 500 experiments allowed the selection of an acceptable spatial clustering method. The results were evaluated in terms of the BSS/TSS indicator and visually in terms of the configuration of the areas of crime, as well as the extent to which they corresponded to the knowledge of the specificities of crime in particular urban areas.

It can be asserted that the crime event data aggregated in small cells (500 m × 500 m in this study) are suitable for the task of city crime regionalisation. The variables used for regionalisation in this study, i.e., the number of the three types of events (VIO, PRO, IPP) separately in open and private/semi private spaces and the population, were found to be appropriate. The best BSS/TSS estimates obtained (BSS/TSS 0.6 to 0.7) are much better than those obtained in previous experiments using all-Lithuania data and cells of up to 5 km$^2$ (BSS/TSS 0.35). However, only with good knowledge of the structure of a particular

city and the general characteristics of crime in the city can conclusions be drawn about the validity and value of regionalisation results.

The examples presented in the study come from experiments in three specific cities. They were of medium size, ranging from 0.17 to 0.62 million inhabitants, with areas ranging from 289 to 737 square kilometres. In the case of crime information, where the variability in parameter values in the grid is high and they are not normally distributed, a better result was obtained by dividing the values into five ranges according to quantiles, categorizing them and separately treating the cells with no recorded criminal events. In general, two things are important in regionalisation: the relative number of events and the structure (ratio of the number of different types of events), which should not be too highly variable. In cases where the areas with the highest population density are very heterogeneous in socio-demographic terms, it is appropriate to have a larger number of intervals with high population density.

For smaller towns, the absolute values of the variables could be used, and additional variables could also be included if they are known to be important in the structure of crime, such as alcohol and drug-related incidents and economic crime (especially in towns near the state border). Additional experiments would be needed to determine the appropriate number of clusters. The maps can be refined by introducing additional regionalisation variables (socio-demographic characteristics and parameters of urban structures).

Out of the several spatial clustering methods that were applied experimentally, the original AZP method produced the best results. Only AZP *Simulated Annealing* generated two clusters in the central part of Vilnius, which would better fit the crime structure of the city, but the other districts extracted via this method were more difficult to explain.

The SCHC and REDCAP algorithms also produced relatively good statistical estimates and similar regionalisation results. The aggregated data used, with 6–8 clusters, were not sensitive to the limitations of minimal cluster size. As expected, Ward's linkage yielded better results than other (min. max, average) linkage methods.

The statistical estimate of the BSS/TSS may increase slightly upon increasing restrictions on clusters or the number of clusters, but in practical terms, a slightly better estimate does not always mean a better result. For example, when the number of cells is large, increasing the minimum cluster size parameter results in clusters that, although homogeneous, are very complex in terms of their configuration. Once sufficiently good BSS/TSS estimates have been obtained (in our case 0.6 or more), the most appropriate number and, if necessary, size of clusters can only be determined based on city-specific knowledge and further experimentation.

In this study, seven areas of crime in Vilnius, Kaunas and Klaipėda have been identified for further analysis. In the city maps, the areas of the same type are shown in the same colours according to the highest similarity, but it should be borne in mind that cities have their own specificities, the most important aspects of which are listed in Table 8.

Regionalisation maps have a different purpose from analytical maps. They are more generalised, covering the structural characteristics of crime in relatively large areas, and are, therefore, more suitable for situational assessment and strategic decision-making by local authorities. Such maps are relatively easy to understand for policy makers or investors who do not have the time and expertise to consider the nuances of the territorial dispersion of crime but need to assess the overall crime situation in a city. Maps of crime areas also help to effectively present the crime situation to urban communities. In public communication, they are less likely to give a false impression of the dispersion of crime (compared, for example, to the hotspot maps used by the experts).

Research into the configuration of crime areas can provide a better understanding of the spatial structure of crime, thus contributing to theoretical insights. In our case, all three cities exhibit somewhat similar spatial patterns in the distribution of crime areas around city centres, but the three examples are too limited to draw more general conclusions. The data used in this study reflect the situation during the COVID-19 pandemic, which is not typical. We plan to compare the results of this study with the regions identified via the

same method for 2022 and beyond ("normal" post-pandemic period). Moreover, the results of the regionalisation presented in this article could be compared to the results of similar studies in similar cities in other countries.

**Author Contributions:** Conceptualisation, Giedrė Beconytė; methodology, Giedrė Beconytė and Kostas Gružas; software and visualisation, Kostas Gružas; validation, Kostas Gružas and Eduardas Spiriajevas; investigation, all authors; writing—original draft preparation, Giedrė Beconytė; writing— review and editing, Kostas Gružas and Eduardas Spiriajevas. All authors have read and agreed to the published version of the manuscript.

**Funding:** This research received no external funding.

**Data Availability Statement:** Research dataset is distributed under CC BY 4.0 at https://atviri-duomenys-vu-lt.hub.arcgis.com/datasets/vu-lt::categorized-police-registered-events-data-of-vilnius-kaunas-and-klaip%C4%97da-in-2020-/about (accessed on 14 December 2023). More crime data and relevant information are available at https://lietuvoskartografija.lt/mokslas-visiems/scientific-publications-projects/spatial-distribution-of-criminal-events-over-lithuania-in-2015-2019 (accessed on 14 December 2023).

**Acknowledgments:** We offer thanks to the Lithuanian Police Department for the provision of crime event data, as well as managers of the national spatial information portal (www.geoportal.lt (accessed on 14 December 2023)) and Statistics Lithuania (www.stat.gov.lt (accessed on 14 December 2023)) for enabling unrestricted access to interoperable data.

**Conflicts of Interest:** The authors declare no conflict of interest.

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
