# Peer review of "Areas of Crime in Cities: Case Study of Lithuania"

_ijgi, doi:10.3390/ijgi13010001_

Round 1
Reviewer 1 Report
Comments and Suggestions for Authors
The authors of the manuscript made a successful attempt to draw crime maps for the three largest Lithuanian cities.
Quite old statistics from the Lithuanian Police on crime were used (for the 2020 year), but these statistics are acceptable. Crime maps were developed for Vilnius, Kaunas and Klaipeda within their administrative boundaries and suburbs. Territorial diversity of crime is shown divided into 7 types of urban areas.
The use of the term “regions of crime” by the authors is questionable. Generally, "region" is a cohesive area that is homogeneous in selected defining criteria and is distinguished from neighboring areas or regions by those criteria. Crime maps in Figures 3-5 show not 7 cohesive “regions of crime”, but 7 types of urban areas in Vilnius, Kaunas, Klaipeda and their suburbs differentiated in terms of crime. It is recommended to apply this remark in the content of the whole manuscript, especially in its topic, objective, research questions, etc.
Author Response
Thank you for your positive evaluation and a discussion. Please see the attached document.
Reviewer 2 Report
Comments and Suggestions for Authors
This paper presents an interesting analysis of crime patterns on Lithuania and it is refreshing to see spatial crime analysis from Eastern Europe were there are fewer studies.
There are some challenges I have with the paper which should be addressed prior to publication.
1) The term regional throws me – although I appreciate this is within cities. Moreover, when you describe regionalisation maps and figure 'd' the ‘regions’ visually look larger than the 500m grids you are using. Indeed I find it hard to consider 500m grids as a region. Or are the grids used in the analysis variable. The methodology followed is vague to follow. Please provide more explicit description of the methods used.
2) The term experiment is also misleading. In criminology this phrase is used to describe experimental policy actions (often with controls) to test whether an intervention works. I suggest simulation is more appropriate given the methods you are using.
3) Please can you explain how the regionalisation maps you are developing are different to the geodemographic classifications produced commonly today – based on a range of clustering methods. The methods you use predominantly rely on clustering techniques. There have been several studies into geodemographics and crime in the literature you should draw from if this is the case.
4) If your method is distinctly different, please explain why – or is similar please explain the use of regionalisation – and how you are defining regions for this study (spatial scale).
5) Your study lacks any theoretical underpinning of crime patterns. For example lines 82-86 you suggest that higher population density increases opportunities for criminal activities – but if this was the case you would only need to model population density and no other characteristics that you describe in Table 8 to explain spatial crime pattens.
6) Why have you used ‘open spaces’ and ‘other’. Reading the descriptions these seem to be private/semi private and open spaces - which would be more commonly used in literature and there are studies here you could draw from.
7) What variables are you using in your modelling/simulations and what was the theoretical basis by which they were selected. Or previous studies these have been drawn from.
8) Line 314 It is not clear how you have carried out the following: ‘Potential crime clusters also can be linked to the socio-demographic and urban territorial structure of the analysed cities’
9) Please provide more details as to the methodology behind table 8.
10) Please provide some further discussion of the advantages of the regionalisation maps over for example analytical maps such as hot spots which are used for hot spot policing. Are these maps more useful for strategic planning/decisions. What insights does this type of analysis have on our understanding of crime (for example Figure 8).
Comments on the Quality of English Language
At times word choice could be improved as I found it hard to interpret the meaning of some sentences.
There are some grammar issues and the sentence structure could be improved to help with the flow of text/reading.
Author Response
Thank you for your detailed and insightful comments. We've done our best to answer in detail and improve the text correspondingly. Please see th eattached file.

Reviewer 3 Report
Comments and Suggestions for Authors
The paper provides an application on crime regions in Lithuanian cities. The paper is well researched based on a systematic protocol and attention directed to better understand the application of crime maps. There is something interesting to find for many researchers interested this kind of techniques.
Advantages:
The work is considered interesting and relevant to the scope of the journal. The researcher understands both the rationale and the methods.
Disadvantages.
1.The summary does not show your main findings well.
2.The introduction needs to better reflect the objective of the work with more clarity and evidence of its main focus. Synthesize in a better way.
3. The graphics should be improved and many of the maps do not have cartographic standards.
4.Point 3 of the document should be called "Results".
5.In the discussion, there is no evidence that they discuss their results with previous studies or confront their work with contributions or problems found in other national or international studies. They should improve the discussion regarding the challenges for a better future implementation of these techniques.
Problems needed to be addressed on the maps:
- Fig. (1): Increase the size of the graphics and improve the description of figures c and d.
- Fig. (2): It does not have, north, scale, coordinates, etc., the location is not explicit, the way in which the idea of location of the cities referring to the country is given should be improved, the white backgrounds are not adequate.
- Table 4 and 6. The colors within the maps to which they refer?
- Fig. (3,4,6): Increase size, this is the result of their analysis should be able to be seen in the best way, place the north, the graphic scale is not visible, increase the size of the scale.
In conclusion, the authors should improve the work in several details to make it more understandable and suitable for publication.
Comments on the Quality of English LanguageImprove punctuation and grammar
Author Response
Thank you very much for the constructive criticism. Please see the attached file for responses.
Round 2
Reviewer 2 Report
Comments and Suggestions for Authors
This revised paper adds needed clarity and provides a much clearer overview of the differences between regionalisation maps and analytical maps. It also draws out the distinctions between geodemographics and the methods used in this study.
The paper should be checked for minor grammatical issues prior to publication but I do not have any further concerns to raise.
Whilst I remain unconvinced about the value of regionalisation maps they do add new insights to the geography of crime literature and align with the purposes of the journal. You have satisfactorily responded to changes I suggested or discussed why they were not appropriate and I am happy to recommend this for publication subject to other reviewer/editor comments.
Comments on the Quality of English LanguageThe paper should be checked for minor grammatical issues prior to publication
Reviewer 3 Report
Comments and Suggestions for Authors
The authors made the suggested changes